# Reversal of Right Ventricular Hypertrophy and Dysfunction by Prostacyclin in a Rat Model of Severe Pulmonary Arterial Hypertension

**DOI:** 10.3390/ijms23105426

**Published:** 2022-05-12

**Authors:** Rebecca R. Vanderpool, Anastasia Gorelova, Yiran Ma, Mohammad Alhamaydeh, Jeffrey Baust, Sruti Shiva, Stevan P. Tofovic, Jian Hu, Seyed Mehdi Nouraie, Mark T. Gladwin, Maryam Sharifi-Sanjani, Imad Al Ghouleh

**Affiliations:** 1Division of Cardiovascular Medicine, The Ohio State University, Columbus, OH 43210, USA; rebecca.vanderpool@osumc.edu; 2Heart, Lung and Blood Vascular Medicine Institute, University of Pittsburgh School of Medicine, Pittsburgh, PA 15213, USA; ana.a.gorelova@gmail.com (A.G.); yim37@pitt.edu (Y.M.); jjb100@pitt.edu (J.B.); sss43@pitt.edu (S.S.); tofovic@pitt.edu (S.P.T.); jian.hu@osumc.edu (J.H.); snouraie@pitt.edu (S.M.N.); gladwinmt@upmc.edu (M.T.G.); 3Department of Pharmacology & Chemical Biology, University of Pittsburgh School of Medicine, Pittsburgh, PA 15213, USA; 4Division of Cardiology, University of Pittsburgh School of Medicine, Pittsburgh, PA 15213, USA; 5Brody School of Medicine, East Carolina University, Greenville, NC 27834, USA; alhamaydehm21@ecu.edu; 6Division of Pulmonary, Allergy & Critical Care Medicine, University of Pittsburgh School of Medicine, Pittsburgh, PA 15213, USA

**Keywords:** pulmonary hypertension, right ventricle, prostacyclin, sugen, hypoxia, fibrosis

## Abstract

Prostacyclin analogs are among the most effective and widely used therapies for pulmonary arterial hypertension (PAH). However, it is unknown whether they also confer protection through right ventricle (RV) myocardio-specific mechanisms. Moreover, the use of prostacyclin analogs in severe models of PAH has not been adequately tested. To further identify underlying responses to prostacyclin, a prostacyclin analogue, treprostinil, was used in a preclinical rat Sugen-chronic hypoxia (SuCH) model of severe PAH that closely resembles the human disease. Male Sprague–Dawley rats were implanted with osmotic pumps containing vehicle or treprostinil, injected concurrently with a bolus of Sugen (SU5416) and exposed to 3-week hypoxia followed by 3-week normoxia. RV function was assessed using pressure–volume loops and hypertrophy by weight assessed. To identify altered mechanisms within the RV, tissue samples were used to perform a custom RNA array analysis, histological staining, and protein and transcript level confirmatory analyses. Treprostinil significantly reduced SuCH-associated RV hypertrophy and decreased the rise in RV systolic pressure, mean pulmonary arterial (mPAP), and right atrial (RAP) pressure. Prostacyclin treatment was associated with improvements in RV stroke work, maximum rate of ventricular pressure change (max dP/dt) and the contractile index, and almost a complete reversal of SuCH-associated increase in RV end-systolic elastance, suggesting the involvement of load-independent improvements in intrinsic RV systolic contractility by prostacyclin treatment. An analysis of the RV tissues showed no changes in cardiac mitochondrial respiration and ATP generation. However, custom RNA array analysis revealed amelioration of SuCH-associated increases in newly identified TBX20 as well as the fibrotic markers collagen1α1 and collagen 3α1 upon treprostinil treatment. Taken together, our data support decreased afterload and load-independent improvements in RV function following prostacyclin administration in severe PAH, and these changes appear to associate with improvements in RV fibrotic responses.

## 1. Introduction

Pulmonary arterial hypertension (PAH) is a devastating disease that is characterized by the aberrant remodeling of cells of the pulmonary vasculature resulting in a chronic elevation of blood pressure in the pulmonary circulation. Right ventricle (RV) dysfunction, remodeling, and failure are a consequence of severe PAH that contributes to 44% of deaths in patients with PAH [1]. Nonetheless, little is known about RV adaptation and failure in severe PAH, and questions remain as to whether or not current therapies targeting the lung circulation can also have a protective effect on the RV. Additional RV-targeted therapies may provide the needed additive protective effect in order to enhance survival.

It is well-established that prostacyclin is protective and among the definitive treatments for PAH that improve survival in patients with advanced PAH [2]. This is believed to mainly occur as a consequence of its vasodilatory, antithrombotic, antiproliferative, and anti-inflammatory effects, which have remained controversial to this date [3,4,5,6]. However, other positive and more profound protective mechanisms, specifically in the RV, cannot be ruled out and have not been fully investigated [7]. Indeed, recent studies provide in vitro and in vivo evidence for the positive RV inotropic effect with prostacyclin treatment [6,8]. Further, available evidence suggests that the pressure-dependent, i.e., mechanical stress on the RV due to elevated mPAP, is not solely responsible for PAH-associated RV failure [9]. Indeed, despite a similar degree of pressure overload, pulmonary artery banding (PAB, similar to pulmonary stenosis induction) in rats is not associated with RV failure [10], while models of pulmonary vascular disease such as Sugen/Hypoxia develop RV failure despite a similar degree of pressure overload [11].

Mitochondrial energetics and metabolic changes are proposed mechanisms contributing to pulmonary vasculature remodeling and RV dysfunction in PAH [12,13,14]. A previous study shows that the RV protective role of prostacyclin involves mitochondrial ATP generation [15]. Meanwhile, the contribution of other tissue-level factors affecting the RV contractile function, such as fibrosis [9,16], remains to be fully studied following prostacyclin treatment. In support of the latter, previous studies show that prostacyclin reduces RV fibrosis and improves RV function in pulmonary artery banding [17], but the precise mechanism(s) by which prostacyclin improves survival in PAH and whether this occur due to an independent effect on the RV is (are) not fully elucidated.

In the present study, our data show that the positive RV effects of prostacyclin treatment in severe PAH may also involve load-independent benefits, evidenced by increased RV contractility along with enhanced end-systolic elastance (Ees). Further, we provide evidence that the amelioration of PAH-associated RV dysfunction with treprostinil treatment involves major pathways relevant to RV fibrosis.

## 2. Results

### 2.1. Treprostinil Ameliorates Pulmonary Arterial and Right Ventricular Dysfunction Mediated by Sugen-Chronic Hypoxia

Sugen-chronic hypoxia (SuCH) increased right ventricular weight (Figure 1A). As appreciated from the pressure–volume loop (Figure 1B), SuCH also increased RV pressure (Figure 1C), which was accompanied by an increase in mean pulmonary arterial pressure (mPAP) and pulmonary vascular resistance (PVR; Figure 1D,E). As anticipated, treprostinil (Trep.) significantly reduced SuCH-induced increases in RV weight (Fulton Index: 0.26 ± 0.01, 0.58 ± 0.04, and 0.37 ± 0.05, for normoxia, SuCH, and SuCH+Trep., respectively) and pressure (mmHg: 24.4 ± 0.7, 77.5 ± 7.9, and 40.5 ± 6.6, for normoxia, SuCH, and SuCH+Trep., respectively). An attenuation in mPAP (mmHg: 16.4 ± 0.5, 77.5 ± 7.9, and 40.5 ± 6.6, *p* < 0.001, for normoxia, SuCH, and SuCH+Trep., respectively) and PVR (mmHg/mL/min: 0.37 ± 0.04, 1 ± 0.18, and 0.48 ± 0.17, for normoxia, SuCH, and SuCH+Trep., respectively) was also observed. Further, the significant increase in RV filling pressure induced by SuCH, as assessed by right atrial pressure (RAP), was reduced by treprostinil (mmHg, RAP: 1.9 ± 0.3, 5.0 ± 0.7, and 3.0 ± 0.3, for normoxia, SuCH, and SuCH+Trep., respectively, Figure 1F).

In terms of RV systolic and diastolic function, treprostinil reversed the SuCH-associated increase in max dP/dt (maximum rate of change in the RV systolic pressure), an indicator of RV contractility (mmHg/sec: 1447 ± 80, 3109 ± 333, and 2022 ± 268, for normoxia, SuCH, and SuCH+Trep., respectively, Figure 2A). Further, a significant improvement in the SuCH-induced decrease in the contractile index (CI; max dP/dt normalized to mPAP: 87.9 ± 3.1, 61.3 ± 3.0, and 77.8 ± 3.9, for normoxia, SuCH, and SuCH+Trep., respectively) was observed, supporting RV function restoration (Figure 2B). We also observed an associated restoration of the SuCH-associated reduction in min dP/dt (minimum rate of RV pressure changes), a measure of RV diastolic relaxation, by treprostinil (mmHg/sec: −861 ± 46, −2378 ± 298, and −1362 ± 252, for normoxia, SuCH, and SuCH+Trep., respectively), evidencing improvement in RV diastolic function (Figure 2C). Further supporting enhanced RV function with treprostinil, there was a reversal of the SuCH-mediated increase of stroke work (mmHg*μL: 2583 ± 350, 9686 ± 877, and 5268 ± 982, for normoxia, SuCH, and SuCH+Trep., respectively, Figure 2D).

We also observed a significant increase in end-systolic (ESV) and end-diastolic (EDV) volumes in response to SuCH (μL: 157 ± 28 and 358 ± 37, respectively, Figure 2E,F), with no change in heart rate (HR; Appendix A) and ejection fraction (EF; Appendix A). Treprostinil treatment resulted in a significant reduction in the SuCH-associated increase in ESV and a non-significant trend toward reduction in EDV (μL: 73 ± 8 and 304 ± 44, respectively, Figure 2E,F) indicative of a dilated RV, but of an enhanced RV contractile function with prostacyclin treatment, as ESV was reduced in the presence of a similar EDV.

Hemodynamic assessment of load-independent cardiac function, which is the intrinsic contractile response of the myocardium, was performed using transient inferior vena cava (IVC) occlusion (a transient reduction in preload) followed by an analysis of the obtained pressure–volume loops in order to calculate end-systolic elastance (Ees). Importantly, the increase in Ees and peak power index induced by SuCH (mmHg: 0.53 ± 0.1 and 212.14 ± 19.32, respectively) were reversed by treprostinil (mmHg: 0.2 ± 0.1 and 113.9 ± 21.3, respectively, Figure 3A,D). Arterial elastance (Ea) was also increased with SuCH and presented a nonsignificant trend toward reduction with treprostinil (mmHg/μL: 0.2 ± 0.002, 0.5 ± 0.09, and 0.2 ± 0.09, for normoxia, SuCH, and SuCH+Trep., respectively, Figure 3B). As such, there were no significant changes to Ees/Ea (SuHx and treprostinil interaction *p* value of 0.5855, Figure 4C), an index of RV–PA coupling, possibly due to both Ees and Ea changing and similar to what we had previously seen [7]. Collectively, these data, together with improved max dP/dt and CI, are supportive of a positive effect of treprostinil on pulmonary vasculature as well as RV intrinsic inotropy, which led to the enhanced RV function, possibly independently and in unison with pulmonary vascular protection.

### 2.2. Beneficial Effect of Treprostinil on the Right Ventricle May Be Independent of Mitochondrial Functional Changes in SuCH Rats

To gain insight into the underlying pathways for the observed hemodynamic changes, we first assessed mitochondrial respiration in isolated RV mitochondria immediately after tissue collection at the end of SuCH and hemodynamic measurements. Mitochondrial respiration using two different substrates of pyruvate/malate or succinate as complex I or complex II substrates, respectively, along with ADP was measured using respirometry as described in the methods section. State 3 respiration showed no significant difference (with either substrate) with treprostinil treatment (Figure 4A,B). Mitochondrial ATP generation (Figure 4C) and hydrogen peroxide (H_2_O_2_) production (Figure 4D) were also assessed in the isolated mitochondria. ATP generation and ROS production were similarly not altered by treprostinil treatment (Figure 4). These data support intact mitochondrial complex function with no detriment in oxidative phosphorylation or synthesis of ATP and ROS production.

### 2.3. SuCH-Activated Fibrotic Pathways in the Right Ventricle Are Attenuated by Treprostinil

To further interrogate possible mechanism(s) of treprostinil-induced improvement of the RV function following SuCH exposure in rats, we custom-designed an RNA array for 128 genes (Appendix A) based on major pathways and effectors involved in ventricular failure, including apoptotic, fibrotic, HIPPO, and mTOR pathways; transcription factors, sex hormone receptor, mitochondria uncoupling/fission, reactive oxygen species, and TGFß and Wnt signaling (heat map, Figure 5). An RNA array analysis of RV tissues with and without SuCH and/or treprostinil was performed. First pass selection of candidate genes was based on the demonstration of at least >1.5 or <0.5-fold change in expression when comparing the normoxia+Veh. vs. SuCH+Veh. groups or the SuCH+Veh. vs. SuCH+Trep. groups. Then, from the narrowed down list, a final selection was made for genes that showed differential gene expression directionality between the two comparison categories (i.e., normoxia+Veh. vs. SuCH+Veh. or SuCH+Veh. vs. SuCH+Trep.). The top genes identified using these criteria and the relevant major pathway/function they correspond to are listed in Table 1. Approximately one third of the identified genes were related to fibrosis. Therefore, we decided to focus on fibrosis-related genes for further validation using transcript-level confirmatory q-PCR experiments (Figure 6). The microarray analysis unveiled a new transcription factor linked to fibrosis, TBX20, that has not been previously studied in the context of PAH and RV failure. The transcript-level analysis confirmed a significant reversal of the SuCH-mediated increase in TBX20 (Figure 6A) upon treprostinil treatment. Interestingly, a cohort of transcription factors were reported to be working with TBX20 to alter gene expression in the adult hearts [18]. However, the role of TBX20 in the PAH setting and RV function remains to be elucidated.

With treprostinil treatment, we also observed a significant reduction in collagen 1α1 and collagen 3α1 (Figure 6B) and TGFß1 (Figure 6C) from their enhanced expression by SuCH. Of note, despite a trend, collagen 1α2 had a nonsignificant increase in expression under SuCH, yet there was a significant reduction in the SuCH+Trep. group compared to SuCH+Veh. Further, collagen 4α was significantly altered by SuCH but not with treprostinil (Figure 6B). Similar observations were made regarding TGFß genes. While TGFß1 mRNA expression was increased by SuCH, there was only a nonsignificant trend with TGFß2 and no change in expression of its receptor TGFßR2 under the same treatment. Only TGFß1 was significantly reversed under treprostinil, with only trends toward a reduction of both TGFß2 and TGFßR2 (Figure 6C). Finally, an increase in the following gene expression was seen in the normoxia+Trep. group: collagen 1α1, collagen 1α2, collagen 3α1, TGFß1, and TGFß2.

To confirm that the elevated transcript levels of collagens were indeed translated into cardiac fibrosis, we stained the RV tissues with Masson’s Trichrome. Tissue fibrosis quantification demonstrated that SuCH indeed induces RV fibrosis, which is alleviated in the SuCH+Trep. group RVs (Figure 7A). Combined with our functional data, these results suggest that the attenuated fibrosis may have contributed to the enhanced RV contractility upon treprostinil treatment (Figure 2 and Figure 3). Both collagens 1 and 3 are major myocardial fibrillar collagens, and it is well established that the tissue inhibitor of matrix metalloproteinase-1 (TIMP1) contributes to the collagen turnover and the remodeling of the ventricular extracellular matrix and that it promotes myocardial fibrosis [19,20]. Therefore, we further analyzed our rat RV tissues for RNA and protein levels of TIMP1. In accordance with elevated TIMP levels in PH patients [21], our PCR and ELISA analyses confirmed that SuCH indeed increases TIMP1 and that this increase is attenuated in the SuCH+Trep. group (Figure 7B,C), confirming a protective role for treprostinil in RV tissue matrix remodeling.

## 3. Discussion

Our results demonstrate that the Sugen/hypoxia (SuCH) rat model of PAH induced increased pulmonary artery pressure, and RV weight and pressure associated with load-dependent and load-independent changes in the RV and contractile function. Treprostinil reduced RV weight and pulmonary artery pressure that was accompanied by reductions in pulmonary vascular resistance and RV stroke work. Compared to the SuCH+vehicle, treprostinil improved RV function with an improvement in RV ejection fraction, stroke work, contractile index, and max dp/dt and reduced RV systolic volume. Reductions in RV preload and min dp/dt also suggest a potential effect of treprostinil on improving RV diastolic function. Further, SuCH induced dilated RV, as evidenced by elevated EDV. However, although EDV was not changed with treprostinil, ESV was decreased in the SuCH+Trep. group, supporting a better RV contractile function. Moreover, whereas a decrease in PVR was observed with treprostinil treatment under SuCH, Ea was not significantly altered, while importantly Ees was significantly reduced. Taken together, these data support that the enhanced RV function by treprostinil may be due to an additional direct effect on the RV that is in parallel to its protective effect on the vasculature. These data support the potential benefit of evaluating treprostinil for novel RV-directed therapies in PAH. Interestingly, in further support of the clinical relevance of these findings, treprostinil decreased the SuCH-mediated cardiac power index, which is an independent predictor of mortality [22,23]. Finally, mitochondrial respiration and gene array analyses in the current study provide evidence that under this model of SuCH, treprostinil does not alter mitochondrial respiration, but it does alleviate RV fibrosis and may do so via the modulation of fibrotic gene expression changes, possibly through TBX20, a novel player in the PAH RV identified in this study.

Largely, in vivo results have been controversial regarding the direct effect of prostacyclin on RV contractility [4,6,24,25], with some suggesting that the beneficial effects of prostacyclin are due to enhanced Ees/Ea coupling and/or a decrease in afterload. Meanwhile, others have shown elevated ventricular contractility with acute prostacyclin infusion [5,6,8] and have also reported that prostacyclin treatment induces a dose-dependent increase in cAMP, the main second messenger essential for both chronotropic and inotropic effects of cardiac contractility, in isolated cardiomyocytes [8]. Further, PAH clinical and preclinical studies reveal that increased mechanical RV pressure overload in response to elevated PVR may not be the sole contributor of PAH-associated RV failure. For example, patients with RV pressure overload associated with pulmonary artery stenosis carry better prognosis than PAH patients [26], arguing for the presence of pressure-independent components of pulmonary vascular disease contributing to the RV remodeling and failure in PAH. On the preclinical side, it is reported that pulmonary stenosis induced by pulmonary artery banding (PAB) is not associated with RV failure [10], while preclinical models of pulmonary vascular disease do present with RV failure, despite having a comparable pressure overload [11]. Confirming the same results, another group shows that chronic progressive RV pressure overload does not lead to severe RV dysfunction and that RV failure in their experimental PAH model is associated with myocardial fibrosis and capillary rarefaction [27]. The same group also shows that PAH-associated RV failure accompanies a decrease in RV tissue VEGF protein levels along with impaired myocardial VEGF transcription and capillary density, which were all attenuated with antioxidant supplementation without any observed changes in lung angioproliferation [27]. These studies along with our current findings emphasize the benefits of an RV-targeted therapy in PAH. Further, expanding on findings from our current work will shed light on the underlying mechanisms and will open the door to RV-centric clinical applications of existing PAH therapies.

For example, future preclinical studies can elucidate the precise molecular mechanisms for the protective effects of treprostinil and other prostacyclins and their analogs on RV fibrosis and contractility. This would allow further understanding of RV pathophysiology and drug action and pave the way for more precise RV-targeted therapies. One may extrapolate also, that, considering our data supporting the parallel effect of treprostinil on the RV, future strategies for the use of prostacyclin analogs may cover the load-independent components of pulmonary vascular disease and extend survival in PAH patients via selectively guiding treprostinil to the RV. Indeed, treprostinil decreases the SuCH-mediated cardiac power index, which is an independent predictor of mortality [22,23]. Clinically, we are also hopeful that our current findings can be extended to investigate the utility of treprostinil as a therapy for other conditions in which RV failure may present and increase mortality, such as left-sided end-stage heart failure and end-stage liver failure, to name a few.

One suggested RV protective role of prostacyclin is the activation of mitochondrial ATP-sensitive K^+^ channels [15], which assists in ATP generative capacity, hence raising the possibility of a positive role for prostacyclin in the improvement of mitochondrial function and possible improvement of RV function (inotropic activity). In search of identifying mechanism(s) of treprostinil-induced improvement of RV function following SuCH exposure, we isolated RV tissue mitochondria from our experimental groups. Interestingly, no apparent changes were observed in mitochondrial respiration and ATP and H_2_O_2_ generation, indicating intact mitochondrial function and supporting the involvement of mitochondria bioenergetics-independent mechanisms in our model. Additionally, our gene array analysis along with our selection and comparison criteria did not reveal any changes in mitochondrial uncoupling or fission-related genes, further supporting data from our isolated mitochondria metabolism studies. Further endorsing this notion, HR is believed to be a main determinant of myocardial oxygen demand, and we do not see any changes among our experimental groups. Nevertheless, the involvement of mitochondrial processes other than respiration, ATP and H_2_O_2_ production, and transcript-level changes in uncoupling and fission proteins in our collected RVs, cannot be excluded. For example, a previous study shows changes in mitochondrial metabolism and fatty acid oxidation in rat RVs from their Sugen/Hypoxia model analyzed right at the end of their 4-week hypoxia incubation [28]. It is important to underline the differences in the PAH models, such as hypoxia duration, and the end point of the studies at which the tissues were collected. Therefore, further longitudinal studies using PAH models at varying stages of the disease are needed to fully elucidate mitochondrial dynamics with disease progression.

Our custom RNA array and PCR analyses revealed TBX20, collagens1α1 and 3α1, and TGFß1 genes to be significantly altered in the RVs of our rat severe PAH model. These genes were, importantly, also reversed by treprostinil treatment. TBX20 has been mainly associated with congenital heart diseases and heart development [29], and mutations in TBX20 have also been found in patients with dilated cardiomyopathy [30]. However, the role of TBX20 in PAH and RV failure has not been studied to our knowledge. Available evidence shows that TBX20 silencing inhibits cell proliferation and cell cycle and induces cell apoptosis in H9C2, HEK293 cells, and mouse cardiomyocytes [29,31] and that conditional genetic ablation of TBX20 in fibroblasts adversely affects cardiac development and postinfarct repair [32]. Interestingly, a cohort of transcription factors are reported to be working with TBX20 in order to alter gene expression in the adult hearts [18]. Therefore, the notion that TBX20 may also contribute to improvement of RV function in PAH, either through regulating expression of genes or via regulating cardiac fibroblast survival and proliferation, becomes attractive and is indeed novel. As such, future in vitro studies in isolated cells, such as cardiomyocytes or cardiac fibroblasts, may bring about further clarification of the exact role of TBX20 in the context of RV dysfunction in PAH and could open an entirely new area of research.

Our observation of alleviated levels of collagens1α1 and 3α1 with prostacyclin is similar to a previous report using a less severe Sugen/hypoxia PAH model where no change in mPAP was observed [17]. The report that RVs from that study’s PAH model exhibited cardiac fibrosis and that this was alleviated with prostacyclin Iloprost without any changes to mPAP further confirms that prostacyclin analogs may have direct effects on cardiac tissue in addition to their effects on the pulmonary vasculature and alleviation of afterload. Our findings add to the previous studies and show that treprostinil has a positive RV contractile effect and that it reverses RV dysfunction associated with severe PAH, at least partly due to the attenuation of RV fibrosis. This link between contractility and fibrosis is based on the notion that although some level of collagen is essential for proper force transmission during systole [33], fibrosis is associated with impaired cardiomyocyte excitation–contraction coupling and cardiac stiffness [34,35], and hence the overall ability of the ventricle to contract. Importantly, considering that antifibrotic therapies that are effective in the left ventricle (pirfenidone and eplerenone) do not have the same effect in the RV [36,37], our finding may bring to light the possible favorable effect of treprostinil for the treatment of RV fibrosis in patients with PAH, adding yet another novel avenue for the repurposing of FDA-approved PAH therapies for dual/multiple use.

It is important to highlight that although treprostinil increases collagen subtype transcripts and TGFß1 at baseline, Masson’s Trichrome staining quantification is not increased in the Trep. experimental group. Further, TIMP1 contributes to the formation of a new extracellular matrix that differs from the healthy matrix and leads to the loss of extracellular matrix integrity [16,38]. However, neither TIMP1 expression nor the contractile function of the Trep. group was different when compared with the Veh. group. Therefore, our results show that the increased collagens 1 and 3 at the transcript level by treprostinil treatment may not manifest in increased myocardial collagen deposition and/or ECM remodeling in the RVs in the absence of a PAH trigger.

Collectively, the current study underlines the effect of treprostinil on load-independent components of pulmonary vascular disease that may directly contribute to improvements of RV function in severe PAH. Further, the role of TBX20 in the RV dysfunction associated with PAH is novel and potentially relevant for future RV-targeted therapies. Our findings shed further light on the cardiopulmonary effects of treprostinil and open the door for future investigation into myocardium-specific protective pathways activated by prostacyclin and its analogs, as well as the future expansion of RV-targeting therapeutic modalities and the possibility for multiuse and concurrent multidirecting of existing PAH therapies.

## 4. Materials and Methods

### 4.1. Animals

All animal experiments were approved by and conducted in accordance with the University of Pittsburgh Institutional Animal Care and Use Committee (IACUC# 13112415). Male Sprague–Dawley rats (250–300 g) were divided into four groups: normoxia plus vehicle, Sugen-hypoxia (SuCH) plus vehicle, normoxia plus treprostinil, and SuHx plus treprostinil (*n* = 8 per group). Animals were implanted with ALZET osmotic pumps (ALZET, DURECT Corp. Cupertino, CA, USA) containing vehicle or treprostinil (900 ng/kg/min; United Therapeutics Corporation, Research Triangle Park, NC, USA). For the normoxia groups, animals were housed in room air (21% O_2_) for 6 weeks. For the SuHx groups, animals were subcutaneously injected at the time of minipump implantation with a bolus of Sugen (SU5416; 20 mg/kg) and housed in hypoxic air (10% O_2_) for 3 weeks followed by 3-week normoxia (21% O_2_). After the 6-week period, invasive hemodynamics were used to assess RV function using pressure–volume loops measured using an admittance pressure–volume catheter (Transonic Systems Inc, Ithaca, NY, USA). Rats were anesthetized using isoflurane and intubated before opening the thorax to expose the heart and lungs. The catheter tip was inserted into the right ventricle to measure right ventricular pressure and volume waveforms. All signals were stored (iox2, EMKA, Falls Church, VA), and later, a postprocess analysis was performed using a combination of commercial analysis software (iox2, EMKA, Falls Church, VA, USA) and custom Matlab (R2014a, MathWorks, MA, USA). Right ventricular function was assessed from steady RV pressure–volume loops (on average 3–4 beats). Quality and accuracy of the RV volume measurements were confirmed by using volume measurements where the minimum phase was in the 2.5–7 range. Measured volumes from animals with a minimum phase outside of the range for the whole study were excluded from the analysis (1 animal). Stroke volume was calculated as the difference between end-diastolic volume and end-systolic volume (SV = EDV–ESV), and the ejection fraction (EF) was calculated as the ratio of stroke volume (SV) to end-diastolic volume. Stoke work was calculated as systolic RV pressure multiplied by stroke volume. Ees was assessed (the slope of end-systolic pressure–volume relationship) using RV pressure–volume loops from inferior vena cava (IVC) occlusion, as a load-independent measure of contractile function.

### 4.2. Tissue

Following hemodynamic parameter measurements, the weight of the whole heart was obtained. The right ventricle was then separated from the left ventricle plus septum (LV+Septum). RV hypertrophy was assessed by the Fulton index (FI; ratio of RV to LV+septum weights). All collected tissues were either flash frozen in liquid nitrogen or fixed with 4% paraformaldehyde at room temperature for further analysis as previously published [39].

### 4.3. Mitochondria Isolation and Function

Cardiac mitochondria were isolated from the RV of each rat by differential centrifugation as previously described [40], and respiration was measured by Clark-type oxygen electrode (YSI 5300A-1; Instech Laboratories, Plymouth Meeting, PA, USA). Briefly, mitochondria were suspended in a buffer composed of RV tissue suspended in a buffer composed of KCl (120 mM), HEPES (10 mM), EGTA (1 mM), KH_2_PO_4_ (5 mM), and sucrose (25 mM) (pH 7.4) and stimulated to respire by the addition of pyruvate (1 mM) and malate (1 mM) or succinate (150 mM) as well as ADP (0.5 mM). Oxygen concentration was recorded using a digital recording device (Dataq, Akron, OH, USA). Respiration rate was adjusted for protein concentration. ATP generation was measured using a luciferin–luciferase-based kit (Invitrogen) as per manufacturer’s instructions. Hydrogen peroxide release by the isolated mitochondria was measured by spectrophotometrically assessing the oxidation of Amplex Red over time.

### 4.4. ELISA Protein Analysis

Rat TIMP1 ELISA kit from abcam (Moston, MA, USA) was used accordingly to the manufacturer’s recommendation. Briefly, 100 μL of each standard and sample was added into each well of a 96-well plate and incubated for 2.5 h at RT while gently shaking. This was followed by 4 times of washing and the addition of 100 μL of Biotinylated TIMP1 Detection Antibody. After 1 h of shaking at RT, the washing step was repeated, and 100 μL of 1X HRP–streptavidin solution was added. The plate was then incubated at RT for 45 min, washed 4 times, and 100 μL of TMB One-Step Substrate Reagent added. After 30 min of room temperature (RT) incubation and the addition of 50 μL of Stop Solution to each well, the plate was immediately read at 450 nm.

### 4.5. mRNA Analysis

Right ventricular tissues were homogenized in Qiazol (Qiagen, Germantown, MD, USA) using handheld homogenizers (Kinematica, avantor, Radnor, PA, USA). This was immediately followed by RNA isolation using Qiagen RNeasy Mini Kit as per the manufacturer’s instructions (Qiagen, Germantown, MD, USA). Isolated RNAs were then used for cDNA synthesis and genomic DNA elimination using RT^2^ Easy First Strand Kit (Qiagen) per manufacturer’s instructions.

### 4.6. RNA Array

Custom RT^2^ Profiler PCR Array was designed to include major cardiac dysfunction pathways and mediators including apoptosis and fibrosis (Table 1 and Appendix A) for gene expression profiling. Plate design was finalized with Qiagen (Germantown, MD, USA). cDNA from collected RV samples and RT^2^ SYBR Green master mix (Qiagen) were added onto the SYBR Green optimized primer loaded plates, and PCR experiment was run on QuantStudio 6 Real-Time PCR System (Applied Biosystems, Waltham, MA, USA). Obtained data were exported using Qiagen online RT2 Profiler PCR Array GeneGlobe.

### 4.7. qPCR

PCR reactions for identified genes were performed on QuantStudio 6 Real-Time PCR System (Applied Biosystems, Waltham, MA, USA) using TaqMan Fast Universal PCR Master Mix, according to the manufacturer’s recommendation (ThermoFisher Scientific, Applied Biosystems, Waltham, MA, USA). Thermal cycling conditions were 95.0 °C for 0.20 h, 95.0 °C for 0.01 h, and 60.0 °C for 0.20 h. ΔΔCt values were obtained by normalizing to the housekeeping gene, heat shock protein 90 (HSP90), and control normoxia and used for quantification of relative expression level of identified genes.

### 4.8. Histological Analysis

Fixed RV tissues were OCT-embedded, cryosectioned, and stained with Masson’s Trichrome as previously published [41]. Briefly, tissue sections were treated with hematoxylin for 5 min, followed by a rinse and 60 min treatment with Masson’s Trichrome. This was followed by 0.5% acetic acid, dehydrated to xylene and cover slipped. Fibrosis level was measured as percentage of fibrotic area to total tissue area using NIS-Element image-analysis software. General Analysis 3 (GA3) and RGB thresholding were used to accurately distinguish the blue areas from the purple areas and to calculate the cumulative area of each.

### 4.9. Statistics

Data are expressed as mean ± standard error mean (SEM). Statistical significance tests were performed using two-way and one-way ANOVA with multiple comparison tests using GraphPad Prism (La Jolla, CA, USA). A *p*-value of <0.05 counted as statistically significant.

## Figures and Tables

**Figure 1 ijms-23-05426-f001:**
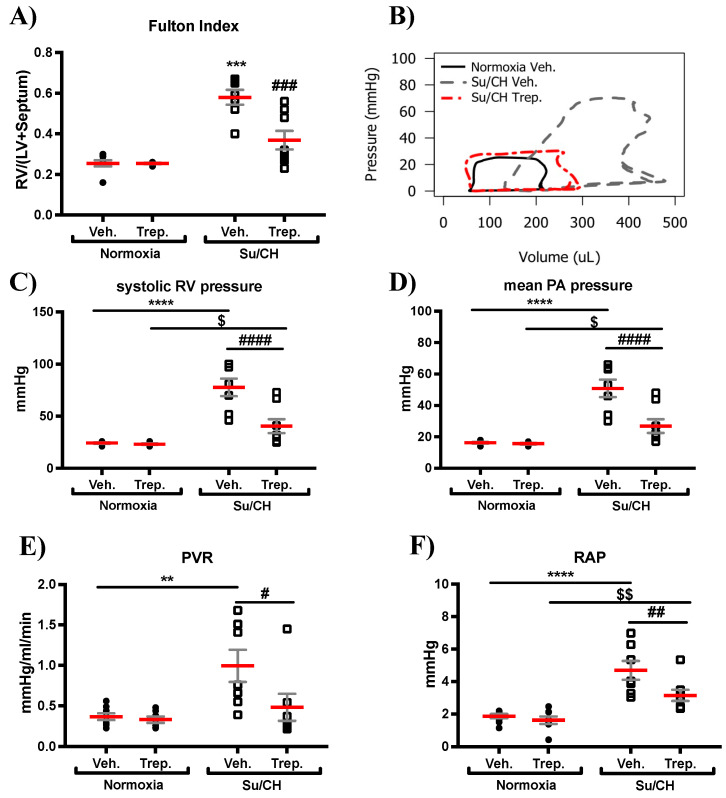
**Hemodynamic assessment of PAH induced by Sugen and chronic hypoxia (SuCH).** Individual value plots for (**A**) Fulton Index. (**B**) representative pressure–volume loops for normoxia vehicle (normoxia+Veh.; black line), SuCH Vehicle (SuCH+Veh.; dotted gray line), and SuCH treprostinil (SuCH+Trep.; dotted red line). (**C**–**F**) grouped individual value plots for (**C**) RV pressure, (**D**) mean pulmonary artery (PA) pressure, (**E**) pulmonary vascular resistance (PVR), and (**F**) right atrial pressure (RAP); *n* = 7–8 animals/experimental group; *, $, and # present significance vs. normoxia+Veh., normoxia+Trep., and SuCH+Veh., respectively; $ and # *p* ≤ 0.05; **, $$, and ## *p* ≤ 0.01; *** and ### *p* ≤ 0.001; **** and #### *p* ≤ 0.0001.

**Figure 2 ijms-23-05426-f002:**
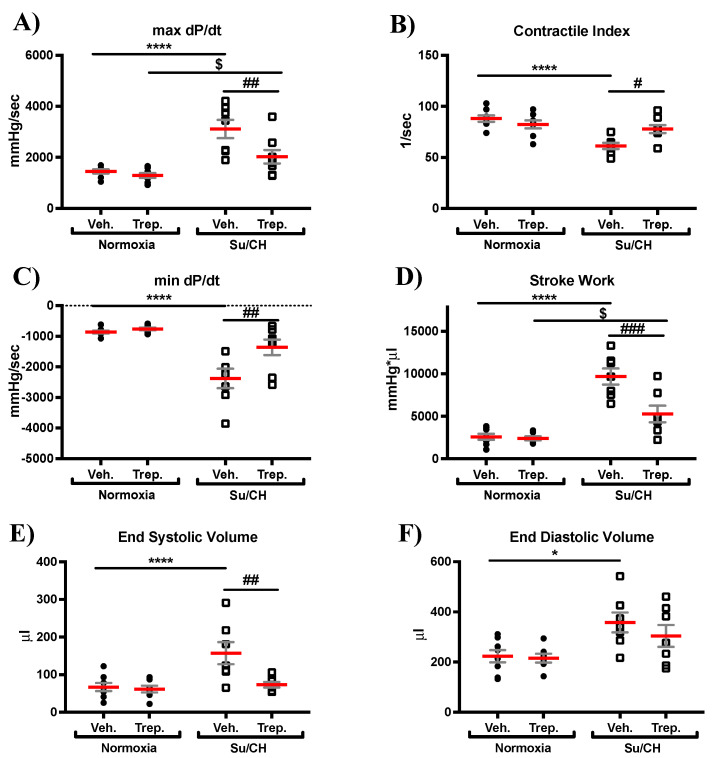
**Development of right ventricular contractile dysfunction with SuCH and its attenuation with treprostinil treatment.** Individual value grouped plots for (**A**) maximum rate of pressure change (max dp/dt), (**B**) contractile index, (**C**) minimum rate of pressure change (min dp/dt), (**D**) stroke work, (**E**) end-systolic volume, and (**F**) end-diastolic volume; *n* = 7–8 animals/experimental group; *, $, and # present significance vs. normoxia+Veh., normoxia+Trep., and SuCH+Veh., respectively; *, $, and # *p* ≤ 0.05; ## *p* ≤ 0.01; ### *p* ≤ 0.001; **** *p* ≤ 0.0001.

**Figure 3 ijms-23-05426-f003:**
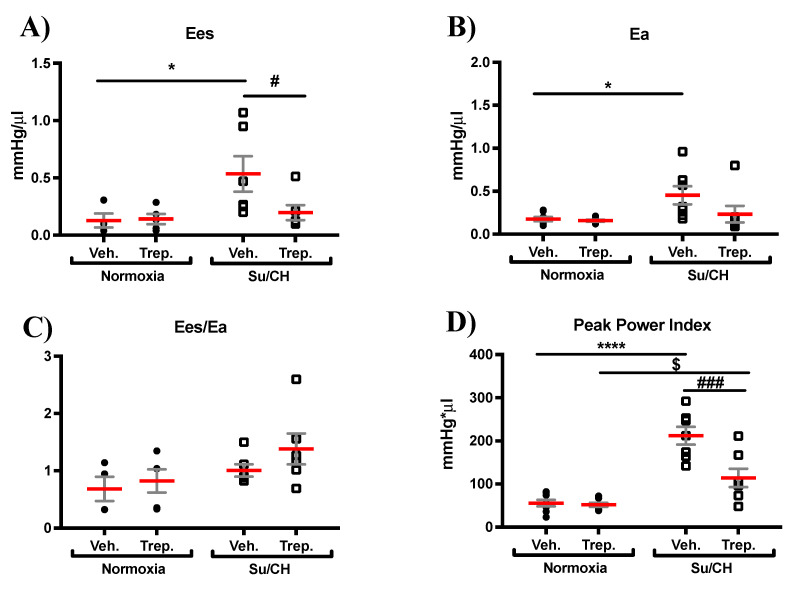
**Effect of treprostinil treatment on load-independent RV function.** Individual value grouped plots for (**A**) end-systolic elastance (Ees), marker of load-independent contractility, (**B**) arterial elastance (Ea), a measure of arterial load, (**C**) Ees/Ea ratio, a marker of RV-PA coupling, and (**D**) peak power index; *n* = 7–8 animals/experimental group; * and # present significance vs. normoxia+Veh. and SuCH+Veh., respectively; *, # and $ *p* ≤ 0.05; ### *p* ≤ 0.001; **** *p* ≤ 0.0001.

**Figure 4 ijms-23-05426-f004:**
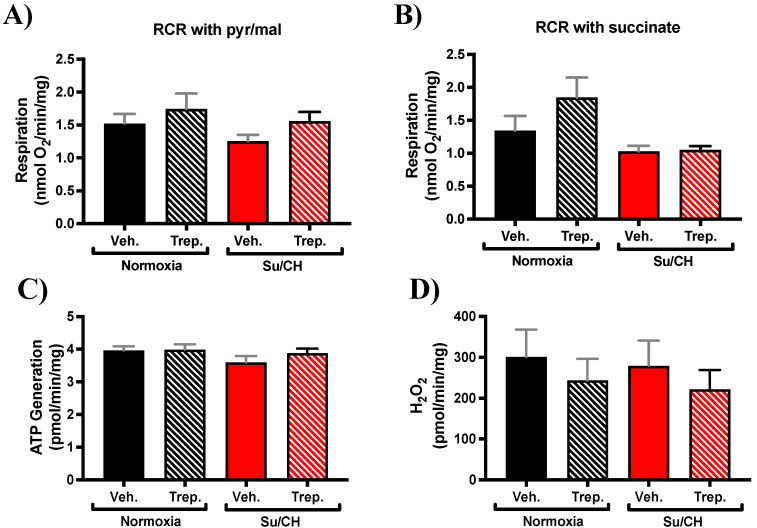
**RV mitochondrial function in PAH rat model of SuCH.** Bar graphs present isolated RV mitochondria respiratory rate in the presence of (**A**) pyruvate/malate or (**B**) succinate as substrates with ADP, (**C**) ATP generation, and (**D**) hydrogen peroxide (H_2_O_2_) production; *n* = 3–8 animals/experimental group.

**Figure 5 ijms-23-05426-f005:**
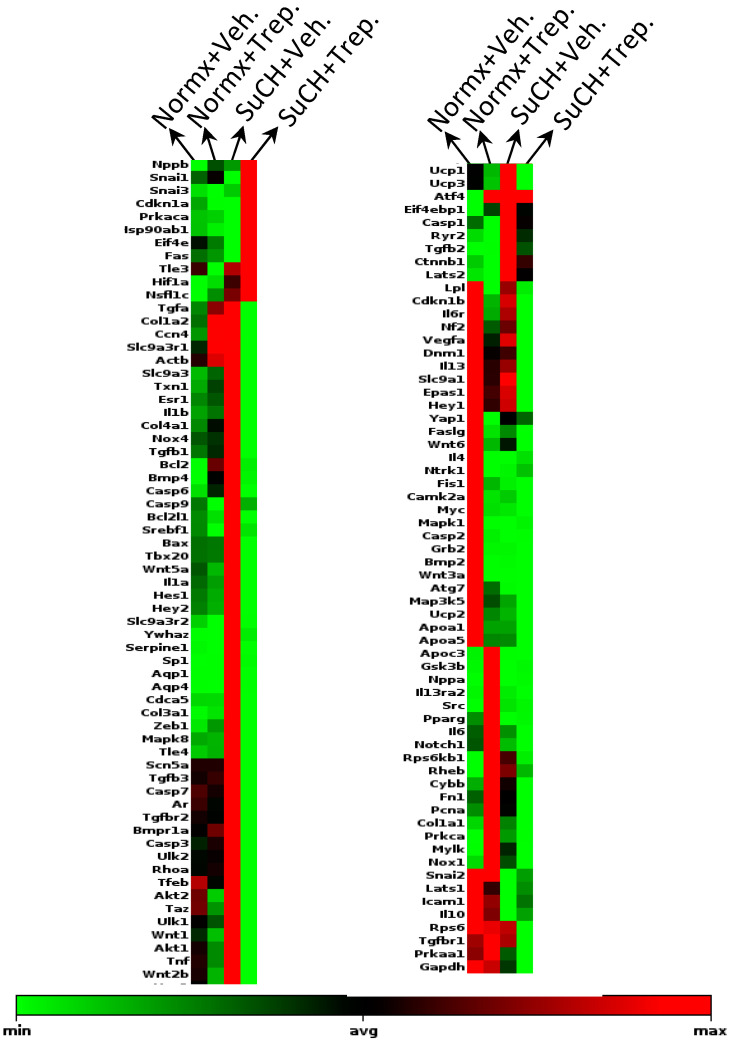
**Clustergram of genes affected by SuCH and treprostinil treatment.** Heat map of gene expression changes in normoxia+vehicle. (normx+Veh.), normoxia+treprostinil (normx+Trep.), SuCH+vehicle (SuCH+Veh.), and SuCH+treprostinil (SuCH+Trep.); *n* = 3 animals/experimental group; green and red represent the minimum and maximum gene expression, respectively, normalized to housekeeping gene HSP90.

**Figure 6 ijms-23-05426-f006:**
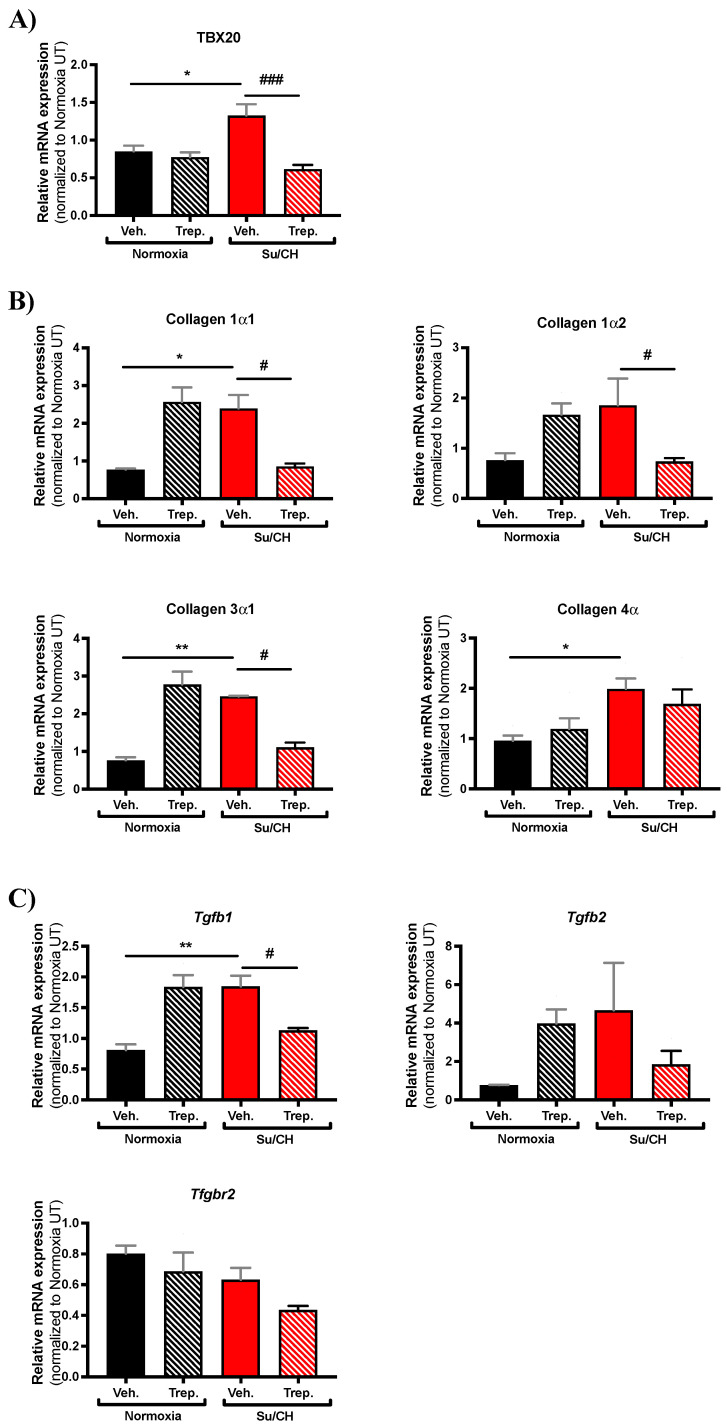
**Expression change assessment of genes identified through custom RNA array.** Bar graphs present quantified transcript levels of (**A**) Tbx20, (**B**) collagen subtypes as identified, and (**C**) TGF-β1, TGF-β2, and TGF-β2 receptor (Tgfbr2) normalized to normoxia+Veh.; *n* = 3–5 animals/experimental group. * and # present significance vs. normoxia+Veh. and SuCH+Veh., respectively; * and # *p* ≤ 0.05; ** *p* ≤ 0.01; ### *p* ≤ 0.001.

**Figure 7 ijms-23-05426-f007:**
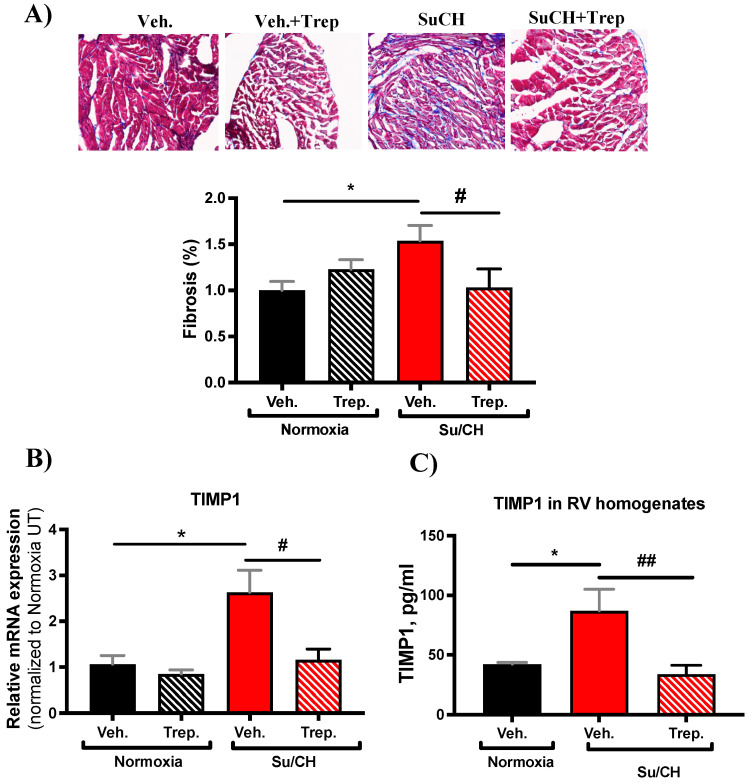
**Outcome of treprostinil treatment on RV fibrosis.** (**A**) Representative Masson’s Trichrome-stained RV tissue images and bar graph quantification of fibrosis (% fibrotic area/ total area normalized to normoxia+Veh.); (**B**) bar graph presenting transcript level of TIMP1 normalized to normoxia+Veh.; and (**C**) bar graph presenting TIMP1 protein-level quantification in RV tissue homogenates by ELISA; *n* = 4–5 animals/experimental group; * and # present significance vs. normoxia+Veh. and SuCH+Veh., respectively; * and # *p* ≤ 0.05; ## *p* ≤ 0.01.

**Table 1 ijms-23-05426-t001:** Top identified genes from RNA array analysis.

Gene	Signaling and Function
Aqp4	Ion channel/hypertrophy
Ctnnb1	Transcription factor
Bcl2l1	Apoptosis
Casp6	Apoptosis
Eif4ebp1	Translation repressor
Zeb1	Transcription factor
Col1a1	Fibrosis
Col1a2	Fibrosis
Col3a1	Fibrosis
Esr1	Sex hormone receptor
Akt1	Fibrosis
IL13ra2	Inflammation
IL4	Inflammation
Rhoa	Signal transduction
Tbx20	Transcription factor/Fibrosis
Tgfb1	Fibrosis
Tgfb2	Fibrosis
Tgfbr2	Fibrosis
Colg4a1	Fibrosis
Tle4	Transcription repressor
Txn1	Redox/Transcription
Ywhaz	Yap signaling

## Data Availability

Not applicable.

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
