# Peer review of "Reversal of Right Ventricular Hypertrophy and Dysfunction by Prostacyclin in a Rat Model of Severe Pulmonary Arterial Hypertension"

_ijms, 2022, doi:10.3390/ijms23105426_

Round 1
Reviewer 1 Report
The paper is well written.
This is an important, unicentric, although with authors from two University Centers in the USA, well designed, an elegant study showing that
reprostinil, a prostacyclin analog that is among the most effective and widely used therapies for pulmonary arterial hypertension (PAH), also benefits the right ventricle (RV) in an experimental model of pre-clinical rat Sugen-chronic hypoxia (SuCH) model of severe PAH that closely resembles the PAH human disease. The methods used are adequate for evaluation of the RV from a functional viewpoint, with assessing pressure-volume loops, and also aimed to identify altered mechanisms within the RV, with RV samples that were used to perform a custom RNA array analysis, histological staining, and further analyzes for protein and transcript levels. On a translational basis, this ideally could help to better understand the patient's symptoms and prognosis when developing RV failure in the reckoning of the PAH. The study is well designed and well-illustrated. The references are appropriate.The Authors have to be commended for their contribution to clarifying prostacyclin analogs use in another context tool for more correct and more comprehensive indications in PAH. I have only a minor suggestion that it would be very much useful to know about future studies that the authors suggest in the field and also, if possible, how the authors see the present results of this experimental study being applied to the clinical area. Best regards,Author Response
Please see the attachment.

Reviewer 2 Report
This manuscript entitled "Reversal of Right Ventricular Hypertrophy and Dysfunction by Prostacyclin in a Rat Model of Severe Pulmonary Arterial Hypertension" by Vanderpool et al tries to elucidate the load-independent effect of prostacyclin on the RV in PAH.
This is a well-written excellent report, and I approve of its publication with minor modifications.
1: Please put a full name for IVC in line 140
2: In line 148, the author wrote "no significant change." Please put a p-value.
3: There are several typo/errors. Please check those carefully. Below are the examples.
Line96: <0.001 <0.001
many figures: ml. should be ul.
Line 409: Check the unit
